# Effects of Isoflavone-Rich NADES Extract of *Pueraria lobata* Roots and Astaxanthin-Rich *Phaffia rhodozyma* Extract on Prostate Carcinogenesis in Rats

**DOI:** 10.3390/plants12030564

**Published:** 2023-01-26

**Authors:** Alexander L. Semenov, Margarita L. Tyndyk, Julia D. Von, Elena D. Ermakova, Anastasia A. Dorofeeva, Irina A. Tumanyan, Ekaterina A. Radetskaya, Maria N. Yurova, Alexander Zherebker, Alexander Yu. Gorbunov, Elena I. Fedoros, Andrey V. Panchenko, Vladimir N. Anisimov

**Affiliations:** 1N.N. Petrov National Medical Research Center of Oncology, 197758 Saint Petersburg, Russia; 2Institute of Biomedical Systems and Biotechnology, Peter the Great St. Petersburg Polytechnic University, 195251 Saint Petersburg, Russia; 3Skolkovo Institute of Science and Technology, 121205 Moscow, Russia; 4Research Institute of Hygiene, Occupational Pathology and Human Ecology, 188663 Saint Petersburg, Russia

**Keywords:** prostate cancer, testosterone, MNU, NADES, *Pueraria lobata*, *Phaffia rhodozyma*, isoflavones, astaxanthin, magnesium, calcium, oxidative stress, rats, FTICR-MS

## Abstract

Prostate cancer (PCa) is one of the most common male malignancies worldwide. In the current study, we evaluated the effects of a natural deep eutectic solvent (NADES) extract of *Pueraria lobata* roots rich in isoflavones (ISF) and *Phaffia rhodozyma* extract rich in astaxanthin (ASX) on an *N*-methyl-*N*-nitrosourea plus testosterone PCa model in rats. ISF consisted of puerarin, daidzein, genistein, formononetin and other polyphenols, while ASX contained lipids and unsaturated species in addition to astaxanthin. Extracts were administered through a whole promotion period in daily doses shown by our group to successfully inhibit benign prostate hyperplasia (BPH) development — 200 mg/kg for ISF and 25 mg/kg for ASX. Though a similar effect was found for BPH processes accompanying PCa induction, the incidence of PCa in animals treated with placebo, ISF and ASX was 37%, 37% and 41%, respectively, showing no chemopreventive activity of ISF and ASX. PCa development was associated with a decrease in the Ca/Mg ratio in serum and an increase in prostate tissue. Treatment with both extracts produced a normalization effect on Ca balance in serum, which, combined with a decrease in the prostatic index, suggests some positive health effects of ISF and ASX.

## 1. Introduction

Among male malignancies, prostate cancer is one of the most common worldwide [1]. As prostate cancer risk is strongly correlated with age [2] and men’s longevity increased throughout the 20th century [3], more of the male population is at risk of the disease. As prostate cancer occurs mostly in elderly men, effective chemopreventive regimens could have a dramatic impact on its morbidity and mortality. There is almost no evidence on how to prevent prostate cancer, but the risk may be reduced by limiting high-fat foods, increasing the intake of vegetables and fruits and performing more exercise [2]. The development of pharmacological or nutritional interventions in prostate cancer prevention is of high interest. It has been reported that a large number of phytochemicals have anticancer properties due to their free radical scavenging activity [4]. Previously, we observed the prominent protective effects of NADES extract of *Pueraria lobata* roots rich in isoflavones (ISF) and *Phaffia rhodozyma* extract rich in astaxanthin (ASX) against benign prostatic hyperplasia development in a rat model [5]. Isoflavones have been identified as dietary components having an important role in prostate cancer with inconsistent and controversial results [6,7,8]. Astaxanthin was shown to decrease the growth of human prostatic cancer cells in vitro [9] and other cancer cell lines [10]. These effects could be due to the complex nature of the studied preparations. One of the promising methods for analyzing such complex substances is Fourier transform ion cyclotron resonance mass spectrometry (FTICR-MS). Its advantage over conventional mass spectrometry is its ability to obtain the exact ratio of mass to charge and, consequently, the accurate masses of the components [11].

This background motivated us to evaluate the chemopreventive effect of isoflavones extracted from *Pueraria lobata* roots and astaxanthin extract from *Phaffia rhodozyma* in an orthotopic rat model of prostate cancer and to assess the applicability of the FTICR-MS technique in the study of such complex substances. To our knowledge, this is the first time the effects of isoflavone-rich NADES extract from *Pueraria lobata* roots and astaxanthin in any form were assessed in orthotopic carcinogen-induced prostate cancer in vivo model. 

## 2. Results

### 2.1. FTICR-MS Molecular Characterization of Isoflavone-Rich NADES Extract of Pueraria Lobata Roots and Astaxanthin-Rich Phaffia Rhodozyma Extract

Both extracts were characterized by rather complex mass spectra with resolved 2123 and 1374 molecular formulae for ASX and ISF, respectively. The scatter plot of the double-bond equivalent (DBE) versus the number of carbon atoms (C) revealed a similar distribution of components from low-molecular-weight compounds saturated with DBE < 5 to high-molecular-weight compounds with DBE exceeding 25 (Appendix A). However, the relative intensity distributions were distinct. ASX was dominated by two groups of compounds: saturated with DBE < 5 with C~20 and C~35. ISF was characterized by the dominance of one unsaturated component with the molecular composition C_21_H_20_O_9_ (*m/z* 415.10348). Likely, this component was puerarin (see [5]). However, other isoflavones were also detected: daidzein (C_15_H_10_O_4_, *m/z* 253.05067), genistein (C_15_H_10_O_5_, *m/z* 269.04556) and formononetin (C_16_H_12_O_4_, low-intensity *m/z* 267.06592). It was also of interest to evaluate the contribution of different chemical classes in the samples under study. The results are represented in Appendix A. The corresponding histogram highlights differences between samples. ISF was highly populated by aromatic species, which likely correspond to various polyphenols. ASX was mostly composed of lipids and unsaturated species, which, according to AI_con_ [12], is unlikely to possess an aromatic ring, as expected for this sample.

### 2.2. Survival Data and Long-Term Toxicity of Isoflavone-Rich NADES Extract of Pueraria Lobata Roots and Astaxanthin-Rich Phaffia Rhodozyma Extract

The first tumor-related death was registered in the prostate cancer (PCa) group on the 203rd day of the experiment. Animals that survived beyond this date were considered to be effective. Cancer induction resulted in decreased survival in the PCa (*p* = 0.0063 vs. intact control (IC, log-rank test) and the PCa + ASX (*p* = 0.0005 vs. IC, log-rank test, Figure 1). The survival curve of the PCa + ISF group did not differ significantly from either the PCa or IC groups (*p* = 0.1941 vs. PCa, *p* = 0.1056 vs. IC, both log-rank test, Figure 1). By the end of the experiment, the PCa + ASX group had reached a 50% survival rate (361 days).

Animals treated with extracts of either ISF or ASX showed no obvious clinical signs of long-term toxicity. All groups that underwent PCa induction had significantly lower body weight and correspondingly higher relative weight of internal organs at the end of the experiment (Table 1), which is apparently associated with exposure to a constant high level of exogenous testosterone. No significant differences were found between the three groups with induced PCa.

PCa induction resulted in mild granulocytosis in all three cancer groups (5.1–5.8 × 10^9^/L vs. 3.5 × 10^9^/L in the IC group, *q* < 0.05) and thrombocytosis in the PCa group (615 ± 42 × 10^9^/L vs. 791 ± 42 × 10^9^/L in IC, *q* < 0.001) while both isoflavone-rich and astaxanthin-rich extracts normalized latter parameters. Other blood counts (Appendix A) and blood serum biochemistry (Appendix A) parameters did not differ between any of the study groups.

### 2.3. Prostate Cancer Incidence, Prostate Index and Urinary Retention

The number of effective animals is shown in Table 2. Three, twelve, seven and thirteen rats were euthanized due to moribund state before the study’s termination in the IC, PCa, PCa + ISF and PCa + ASX groups, respectively. Cases of PCa were absent in the IC group. In animals with modeled PCa, its frequency was significantly increased and reached 37%, 37% and 41% when counted for the whole prostate in the animals treated with the placebo, ISF and ASX extracts, respectively (Table 2). Cancers that were observed in the dorsolateral prostate occurred in 30% of PCa rats, 33% in animals in the PCa + ISF group and 31% in rats in the PCa + ASX; these differences were not significant. Prostate cancer was less frequent in the ventral lobes and absent in the anterior lobes. Cases of prostatic intraepithelial neoplasia were absent in control animals and accounted for only 7% of animals in each of the induced PCa groups regardless of treatment.

Somatic indices of the prostate were significantly increased in all groups with induced cancer in both rats without tumors and rats with tumors (Figure 2). A two-factor ANOVA (treatment x tumor status) showed significant effects both for treatment (*p* < 0.001, 73,90% of total variation) and diagnosed prostate cancer (*p* < 0.001, 3,64% of total variation) and a significant interaction between these two factors (*p* = 0.03, 1,927% of total variation). Post-hoc multiple comparisons for the main treatment effect showed a significant difference between the PCa group and groups receiving ISF (*q* = 0.008) and ASX (*q* = 0.0012).

Urinary retention was found in one animal in the IC group (3%), in seven animals in the PCa group (23%, *p* = 0.0261 compared to IC) and in eight animals in the PCa + ASX group (28%, *p* = 0.011 compared to IC). When treated with isoflavone-rich extract, urinary retention was observed only in three animals (11%, not significantly different both from the PCa (*p* = 0.304) and the IC groups (*p* = 0.329)).

### 2.4. Assessment of Androgen Receptor Expression and Proliferation

There was no significant difference in the ratio of AR-positive cells in all groups (Figure 3). The number of p-Ac-Histone H3-positive cells (proliferation marker) increased under PCa induction (Figure 3). Treatment with both ISF and ASX tended to decrease the ratio of p-Ac-Histone H3-positive cells (ANOVA all-group comparison *p* = 0.0979, Figure 3).

### 2.5. Prostate Tissue Biochemical Analysis

Superoxide dismutase (SOD) activity significantly decreased in all experimental groups compared to the IC (Table 3). PCa induction resulted in a marked shift in the calcium to magnesium (Ca/Mg) ratio in all experimental groups due to the increase in Ca content. Treatment with the ASX resulted in a statistically significant decrease in Ca content compared to the PCa group, although it did not result in a pronounced change in the Ca/Mg ratio.

### 2.6. Blood Biochemical Analysis

PCa induction was associated with a decreased PSA level in blood serum (Table 4), while treatments with extracts rich in isoflavones or astaxanthin resulted in lower PSA levels compared to the PCa group. Testosterone and 5-alpha-dihydrotestosterone (DHT) levels were significantly higher in the PCa groups compared to the IC group, with a significantly higher level in astaxanthin-extract-treated animals compared to the PCa placebo. At the end of the study, no significant differences were found between the groups in terms of protein metabolism and markers of liver and kidney function. For more details, please refer to the (Appendix A). Cholesterol levels were decreased in all experimental groups compared to the IC (Table 4).

PCa induction caused a marked decrease in serum Ca levels resulting in decreased Ca/Mg ratio. Treatment with both ISF and ASX contributed to the normalization of the Ca/Mg ratio by increasing the Ca content. Isoflavone-rich extract also enhanced serum phosphorus (P) content, which decreased in the PCa and PCa + ASX groups compared to the IC group. PCa induction was also accompanied by a decrease in SOD activity and an increase in malondialdehyde (MDA) levels (Table 4). Treatment with ISF resulted in a more pronounced pro-oxidant effect as compared to the PCa placebo group.

## 3. Discussion

The FTICR-MS results coincided with a previously performed HPLC analysis of isoflavones preparation, with puerarin being the main component in the mix. Our results show the high complexity of the molecular components and classes of the studied preparations, which points to the importance of assessing the possible influence of minor components on the main active ingredients’ effects. It is worth noting that the high complexity of the studied samples has been revealed by FTICR-MS applied in an untargeted way, which became an accepted strategy for annotating plant metabolites [13]. Despite the structures of metabolites being missing, the application of FTICR-MS possesses an advantage over conventional LC with tandem mass-spectrometry, which is limited by the abundance of ions resulting in the loss of two-thirds of possible important metabolites [14]. At the same time, FTICR MS enables an assignment of exact elemental composition to these species.

The rat model used in this study is recognized as reliable for human prostate cancer [15] and was able to produce cancer. The increased PSA level and prostate somatic index are consistent with other reports [16]. Treatment with the NADES extract of isoflavones from *Pueraria lobata* roots or with the extract of astaxanthin from *Phaffia rhodozyma* did not show a favorable effect on the carcinogenesis process. The two treatments did not affect prostate cancer induction. In the PCa group, total cancer incidence was 37% in the whole gland combined. The incidence in the group treated with ISF was 37%, and in the group that received ASX, it was 41%. The carcinogenic process was accompanied by benign prostatic hyperplasia development, which was evident by somatic indices of the prostate in animals without tumors. Astaxanthin-rich extract from *Phaffia rhodozyma* and isoflavone-rich NADES extract from *Pueraria lobata* roots were able to reduce the development of benign prostatic hyperplasia, consistent with our previous study [5].

The anticancer properties of puerarin, daidzein, genistein and formononetin, the main components of the studied isoflavone-rich NADES extract of *Pueraria lobata* roots, are widely known and have been extensively reviewed [6,17,18,19,20,21]. All four isoflavones have been reported to have anticancer activity. Daidzein and genistein are common soy isoflavones. Total soy consumption in clinical trials was systematically reviewed in a meta-analysis covering 266,699 participants and 21,612 prostate cancer patients and was demonstrated to be associated with a reduction in prostate cancer risk [8]. The protective effect of isoflavones against the development of prostate cancer has been reviewed and is considered to be mediated by hormone-like effects through estrogen receptor α and β (ER-α, ER-β) binding or by non-hormone-like effects, including the inhibition of tyrosine kinases, modulation of cell proliferation, regulation of the cell cycle, apoptosis and angiogenesis [17]. In particular, genistein demonstrates the ability to induce apoptosis in cancer cells via the inhibition of ROS scavengers [22]; the ability of various isoflavones to exhibit both anti- and pro-oxidant effects is also well studied [23,24]. In our study, there was no effect of ISF on the antioxidant status in prostate tissues and a pronounced pro-oxidant effect in blood serum.

Isoflavones may exert an indirect effect on the androgen receptor pathway by downregulating the expression of androgen-dependent genes such as prostate-specific antigens [20]. Nevertheless, in our study, PSA levels in animals treated with ISF were at the same level compared to the PCa group. A decrease in PSA levels in patients with PCa and exposed to isoflavones is not always observed [20]. In addition, some studies even report a stimulating effect of isoflavones on cancer cells. For example, formononetin at low doses promoted the proliferation of nasopharyngeal carcinoma cell line CNE2 [25]. 

The anticancer effect of puerarin is dose-dependent, and both androgen-independent and androgen-dependent prostate cancer cells require different exposure times and concentrations of puerarin in a culture medium to exhibit the desired effect [26]. The association between soy and isoflavones and prostate cancer incidence may differ according to disease stage. For example, a study conducted on 43,580 Japanese men with no history of cancer or cardiovascular disease (aged 45–74 years) followed from 1995 to 2016 suggested that a high intake of soy and isoflavones might increase the risk of prostate cancer mortality [7]. Thus, the isoflavone-rich extract from *Pueraria lobata* roots studied here may lack an effect due to multiple reasons, including the level of dose studied, the combination of different isoflavones, and the model implemented (as per our study protocol, treatment commenced at the promotion stage and lasted through the progression stages).

Astaxanthin’s health effects and anticancer properties are established and have been reviewed mechanistically and systematically [10,27]. Astaxanthin inhibits the proliferation of androgen-independent human prostate DU145 cells at concentrations of 0.1–0.2 mM [28] and the growth of androgen-dependent human prostate LNCaP cells at concentrations starting from 0.1 mM, with a prominent effect at 1 mM [29]. Thus, the chemopreventive effect of astaxanthin may depend on the dose level and the molecular peculiarities of the prostate cancer model. As the protective effect of astaxanthin-rich extract from *Phaffia rhodozyma* against benign prostatic hyperplasia was observed in the study, we hypothesize that dose levels for the chemopreventive effect of astaxanthin against benign prostatic hyperplasia and prostate cancer may differ. The pharmacological effect of astaxanthin-rich extract from *Phaffia rhodozyma* in the current study was evident from the increased testosterone and DHT levels compared to the PCa placebo. Increased testosterone levels might be related to the inhibition of the 5a-reductase enzyme, but DHT levels were also significantly higher than those in the placebo-treated PCa group. The expression of androgen receptors and PSA is known to be up-regulated in response to DHT [30]. However, PSA levels in rats treated with ASX were significantly decreased compared to other groups. In contrast to our previous results in the benign prostate hyperplasia (BPH) model [5] and the background data showing the antioxidant action of astaxanthin [31], the current study did not exert any significant effects. Thus, the astaxanthin effects on testosterone and its metabolism may be rather complex.

We observed decreased serum Ca content in the PCa rats. This finding is consistent with low calcium levels occurring in up to 30% of patients with advanced prostate cancer [32]. Hypocalcemia in patients is primarily related to osteoblastic bone metastases. We did not assess bone damage in the model of PCa used. However, both ISF and ASX normalized Ca levels, bringing the values towards the range in the IC.

Comparison with our previous study [5] of isoflavone-rich and astaxanthin-rich extracts in the BPH model has shown similar changes in the Ca/Mg ratio in both models—decrease in serum and increase in prostate tissues—and similar action from both extracts. On the other hand, some differences in the effects on antioxidant parameters were present. In comparison with the BPH model, SOD activity was diminished to a lesser extent in the PCa rats compared to intact animals, and isoflavone-rich extract exhibited marked pro-oxidant effects in the PCa model.

Thus, despite the lack of chemopreventive activity against PCa in the current study, our results support the positive health-promoting effects of ISF. These include, for example, a tendency to increase survival rate; a significant reduction in the prostate index; a decrease of urinary retention incidence; lower proliferation markers in non-malignant prostate tissues; and normalization of serum P, Ca and Ca/Mg ratio. To a lesser extent, health-promoting effects were observed for animals treated with ASX. A decrease in the prostatic index and the normalization of prostatic and serum Ca and serum Ca/Mg ratio were demonstrated.

Further studies with higher doses and/or other formulations, including other PCa models, are needed to explore the chemopreventive potential of isoflavone-rich NADES extract of *Pueraria lobata* roots and astaxanthin-rich *Phaffia rhodozyma* extract against PCa. In the future, it may be useful to take into account the possible effects of minor components when evaluating the molecular mechanisms of action.

## 4. Materials and Methods

### 4.1. Study Materials

We have used the same sources, extraction technologies, and quality control procedures for the isoflavone-rich NADES extract of *Pueraria lobata* roots and astaxanthin-rich *Phaffia rhodozyma* extract in the previous study using the BPH model [5].

Dried kudzu (*Pueraria lobata*) roots were purchased from Xi'an Sgonek Biological Technology (Shaanxi Sheng, China). The roots were ground to a fine homogeneous powder, and the powder was subjected to extraction with natural deep eutectic solvent (NADES) under ultrasonic (UAE) treatment.

In brief, NADES containing choline chloride and citric acid in a ratio of 1:2 mol/mol were used. Liquid NADES was prepared by adding H_2_O (20 wt%) to the dry reagent mixture and heating to 60–80 °C under constant stirring until a stable and clear liquid was formed (60–90 min). The plant material was treated with 20 parts of NADES, and the fraction containing isoflavones was extracted under ultrasonication at 37 kHz at 60 °C for 180 min. Then the liquid part was separated by centrifugation at 6000 rpm for 10 min, and afterward liquid–liquid extraction of the fraction enriched in isoflavones was performed using ethyl acetate. Final isoflavone-rich NADES extract was concentrated with a rotatory evaporator until complete dryness.

The concentration of total isoflavones in the NADES extract of kudzu roots was 30%, where the major isoflavone compounds were puerarin (27.3%), daidzein (2.5%), genistein (0.04%) and formononetin (0.17%) (as per HPLC data [5]).

Lyophilized cells of *Phaffia rhodozyma* Y1654 were purchased from National Bioresource Center (All-Russian Collection of Industrial Microorganisms, Research Institute of Genetics and Selection of Industrial Microorganisms, Moscow, Russia). Cells were revived and grown on solid Yeast Extract–Peptone–Glucose (YPG) media (containing per liter, 20 g glucose, 10 g peptone, 2 g yeast extract, and 9 g agar) at 20 °C. After 3 days of growth, seed cultures were prepared and were further cultivated in YPG medium.

For astaxanthin extraction, cell suspensions of grown yeast colonies were mixed with DMSO at a ratio of approximately 1:4. The suspension was subjected to ultrasonication at 80 kHz in a water bath (set at 35 °C) for 30 min. The suspension was further treated repeatedly with equal volume of petroleum ether, each time subjecting the mixture to frequent vigorous shaking for 10 min, followed by centrifugation at 4000 rpm to separate the colored petroleum ether. Each time a new portion of ether was used, and the treatment was carried out until the pigment exited into the extractant ceased. The pulled portions of petroleum ether containing astaxanthin were evaporated on a rotary evaporator at 45 °C and 100 rpm. After evaporation, the extract was washed 3 times using distilled water with 20% NaCl (100 mL) added for better phase separation. Final petroleum ether phase was mixed with anhydrous NaSO_4_, filtered and then evaporated until dry.

The studied extract obtained from *P. rhodozyma* contains 76.6% (*w*/*v*) astaxanthin (as per HPLC data [5]).

Production and quality control of extracts used in current study was performed at the Institute of Chemical Technology, Ural Federal University, named after the first President of Russia, B. N. Yeltsin (Yekaterinburg, Russia).

### 4.2. High-Resolution Mass Spectrometry Characterization of Study Materials

Molecular compositions of samples of ISF and ASX extracts were acquired on FT MS solariX XR mass spectrometer (Bruker Daltonics, Billerica, MA, USA) equipped with 7 T superconducting magnet and electrospray mode operated in negative ion mode. Prior to analysis, samples under study were redissolved in methanol to concentration of 10 mg/L. Final solutions were directly injected into the ESI source using a microliter pump at a flow rate of 120 μL/h with a nebulizer gas pressure of 138 kPa and a drying gas pressure of 103 kPa. A source heater temperature of 200 °C was maintained to ensure rapid desolvation in the ionized droplets. The mass spectra were both externally and internally calibrated: the former was performed using sodium trifluoroacetate (TFA, 0.01 mg/mL in H_2_O/MeOH 50/50 *v*/*v*), and the latter was performed by the known residual peaks of fatty acids [33] reaching an accuracy value of <1 ppm. Spectra were acquired with the nominal resolution of 530,000 at *m/z* = 400. Peak lists were obtained using DataAnalysis 4.0 software (Bruker), and the molecular formulae were assigned using open-source UltraMassExplorer application (http://dockersrv1.awi.de:3838/ume, accessed on 22 November 2022) [34]. The generated CHONS formulae were validated applying sensible chemical constraints for negative ESI: (O/C ratio ≤ 1, 0.3 < H/C ratio ≤ 2.2; element counts [C ≤ 120, H ≤ 200, 1 < O ≤ 60, N ≤ 5, S ≤ 1]; and mass accuracy window <1 ppm). Molecular components were further divided into six chemical classes [35] based on constrained aromaticity index AI_con_ [12]: lipids (H/C ≥ 1.5, O/C < 0.3), aliphatics (H/C ≥ 1.5, O/C ≥ 0.3, N = 0), N-containing saturated compounds (H/C ≥ 1.5, O/C ≥ 0.3, N > 0), unsaturated (H/C < 1.5, AI_con_ < 0.5), aromatics (0.5 ≤ AI_con_ < 0.67) and condensed (AI_con_ ≥ 0.67).

### 4.3. Animals

A total of 117 male Wistar rats (“Rappolovo” animal facility, Leningrad Region, Russia) were used in the experiment. The animals weighed 268 ± 28 g at the beginning of the study. The animals were housed in conventional polycarbonate cages 1291H (Tecniplast, Buguggiate, Italy) up to 4 per cage under an artificial 12 h light/dark cycle, temperature at 21–23 °C, average humidity 20–55% and ad libitum access to pelleted laboratory chow (Laborotorkorm Ltd., Moscow, Russia) and tap water.

### 4.4. Experimental Design

Rats were randomized into four groups. Control animals (IC, *n* = 31) were intact and were treated with placebo. The remaining groups of animals were subjected to induction of prostate cancer in a two-stage model as described [36]. Briefly, rats were surgically castrated under inhalation anesthesia, and 21 days later, prostate stimulation with prolonged testosterone (833 mg/kg, Omnadren 250, Jelfa, Poland) was performed. After 3 days at the peak of prostatic cell proliferation, carcinogenesis was initiated by a single intravenous injection of N-methyl-N-nitrosourea (50 mg/kg, Sigma–Aldrich, Burlington, MA, USA). Seven days later, animals were randomly subdivided into experimental groups (was taken as day 0 to assess survival). Carcinogenesis was promoted by injections of Omnadren 250 once a week at a dose of 16.7 mg/kg until the end of the study.

All test substances were dissolved ex tempore in refined sunflower oil and administered by gavage in 0.5 ml of solution per rat. Treatment commenced upon the formation of experimental groups. Animals in the second group (PCa, *n* = 30) were administered daily with placebo (refined sunflower oil). The third group (PCa + ISF, *n* = 27) was exposed daily to isoflavone-rich NADES extract of *Pueraria lobata* roots (200 mg/kg). Group four (PCa + ASX, *n* = 29) was exposed daily to astaxanthin-rich *Phaffia rhodozyma* extract (25 mg/kg). The selected doses were shown to be effective in our previous study of the BPH model in rats [5]. The experiment was terminated 12 months after the start of the treatment. During the study, animals were monitored daily, and if found to be moribund were euthanized by CO_2_ inhalation. At the end of the experiment, the animals were euthanized with CO_2,_ and blood samples were obtained by cardiac puncture for blood biochemistry analysis. All animals were subjected to complete autopsy and subsequent pathomorphological examination.

### 4.5. Pathomorphological Examination

Animals were weighed, autopsied and diagnosed with urinary retention on the grounds that the bladder volume exceeded the normal range of 1.44 ml [37]. The prostate was excised free of adherent tissues. The anterior lobes of the prostate were separated from the seminal vesicles. The dorsolateral and ventral lobes were cleared of adipose tissue, and the bladder was removed. All dissected lobes of the prostate were weighed separately on an analytical balance (A&D HR-150AZG, Tokyo, Japan). Prostate somatic indices were calculated by dividing organ weight (mg) by body weight (g) and multiplying by 100. Part of the dorsolateral lobe was frozen in liquid nitrogen for subsequent analysis.

Tissue samples were fixed in 10% neutral buffered formalin and embedded in paraffin after processing according to the standard technique. Next, 4 μm sections were prepared and stained with hematoxylin-eosin. Microscopy was performed on a Nikon Eclipse Ni-U optical microscope (Nikon Corporation, Tokyo, Japan) with a digital camera and NIS-Elements Br software (version 4.30.00; Nikon Corporation).

Prostate tissue sections were also deparaffinized in xylene and immunohistochemically stained with anti-p-Ac-Histone H3 (sc-56739, Santa Cruz, Dallas, TX, USA) and anti-androgen receptor (ab13327, Abcam, Boston, MA, USA) antibodies and percentage of cell with positive nuclei were counted (ImageJ software, NIH, USA).

### 4.6. Biochemical Analysis, Antioxidant Activity Evaluation, Blood Count and ELISA Assay

Biochemical parameters of blood serum and prostate tissue supernatants were assessed using a Konelab 20 analyzer (Thermo Scientific, Vantaa, Finland). Commercial kits for determination of total protein, cholesterol, triglycerides, glucose, calcium and magnesium (NPC "Eco-Service", St. Petersburg, Russia) were used according to manufacturer's protocols. The activity of the antioxidant enzyme superoxide dismutase (SOD) and the concentration of malondialdehyde (MDA) in erythrocyte lysate and prostate tissue homogenate were evaluated on a Konelab 20 analyzer (Thermo Scientific, Vantaa, Finland) according to the protocol described earlier [38]. SOD activity was expressed as relative activity compared to the IC group, which was taken as 100%. Peripheral blood samples were analyzed on a Mindray BC-2800Vet hematology analyzer (Shenzhen Mindray Bio-Medical Electronics Co., Ltd, Shenzhen, China). Frozen prostate tissue samples were mechanically homogenized, suspended in tris-buffered saline/Polysorbate 20 buffer (1 mL) and centrifuged (12,000× *g* for 20 min, 4 °C). Supernatants were collected for further analysis.

Prostate-specific antigen (PSA; Cloud-Clone Corp., Katy, TX, USA), testosterone (DRG, Marburg, Germany) and 5-alpha-dihydrotestosterone (DHT; DRG, Marburg, Germany) were evaluated by enzyme-linked immunosorbent assay (ELISA) using ELISA kits according to manufacturer's protocols. Optical density was measured on an iMark microplate absorbance reader (BioRad, Hercules, CA, USA).

### 4.7. Ethical Standards

The experiments in the study were performed according to International Guidelines for the Care and Use of Vertebrate Animals (European Council Directive; EU2010/63), and the protocol was approved by the Ethics Committee of the N.N. Petrov National Medical Research Centre of Oncology (St. Petersburg, Russia) (protocol No. 3/215, 22/09/2020).

### 4.8. Statistics

Statistical analysis was performed using GraphPad Prism 8 software. The Anderson–Darling test was used to assess data normality. The significance of differences between groups was tested using two-sided Fisher`s exact test, one-way and two-way ANOVA, or a mixed model test with post-hoc multiple comparisons corrected by controlling false discovery rate (FDR; two-stage linear step-up method of Benjamini, Krieger and Yekuteili and Q set as 0.05). The log-rank test was used to perform the survival analysis and compare the Kaplan–Meier survival curves. Data are presented as mean with SEM.

## 5. Conclusions

The studied substances had a complex composition typical among all plant extracts. Isoflavone-rich NADES extract of *Pueraria lobata* roots contained four previously found HPLC target components with a predominance of puerarin as well as multiple aromatic compounds, most probably corresponding to different polyphenols. Lipids and non-saturated compounds were present in the astaxanthin-rich *Phaffia rhodozyma* extract.

Isoflavone-rich NADES extract of *Pueraria lobata* exerted several health-promoting effects, including a significant reduction in the prostate index, decreased urinary retention incidence, inhibition of non-malignant prostate tissue proliferation and normalization of P and Ca balance but had no chemopreventive effect.

Astaxanthin-rich *Phaffia rhodozyma* extract improved the prostatic index and normalized Ca balance but also failed to demonstrate chemopreventive action against PCa in the model.

## Figures and Tables

**Figure 1 plants-12-00564-f001:**
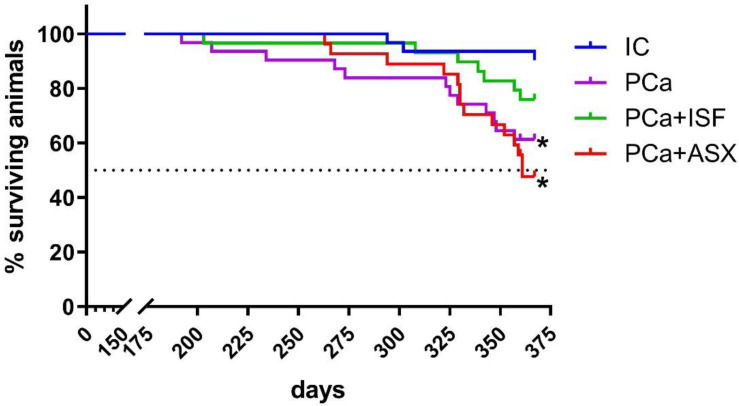
Survival curves of rats with (PCa) and without (IC) induced prostate cancer. ISF—extract of *Pueraria lobata* roots rich in isoflavones. ASX—*Phaffia rhodozyma* extract rich in astaxanthin. *—*p* < 0.05 vs. IC (log-rank test).

**Figure 2 plants-12-00564-f002:**
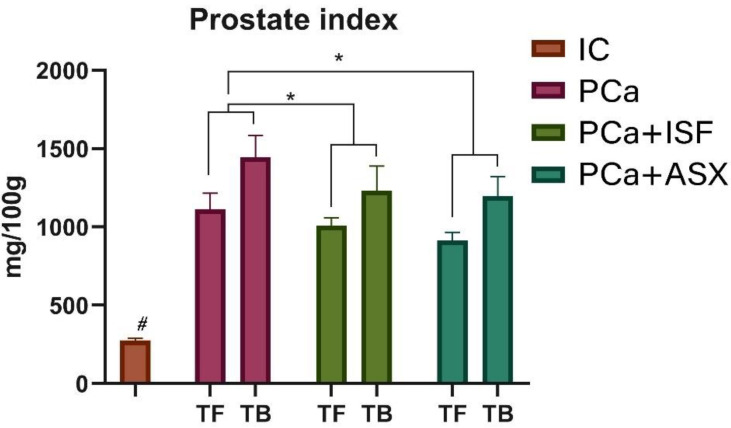
Prostatic index in tumor-free and tumor-bearing animals. IC—intact control. PCa—prostate cancer. ISF—extract of *Pueraria lobata* roots rich in isoflavones. ASX—*Phaffia rhodozyma* extract rich in astaxanthin. TF—tumor free; TB—tumor-bearing. #—all groups with PCa induction were significantly different (*q* < 0.05) from the IC group. *—*q* < 0.05 compared to PCa for treatment factor.

**Figure 3 plants-12-00564-f003:**
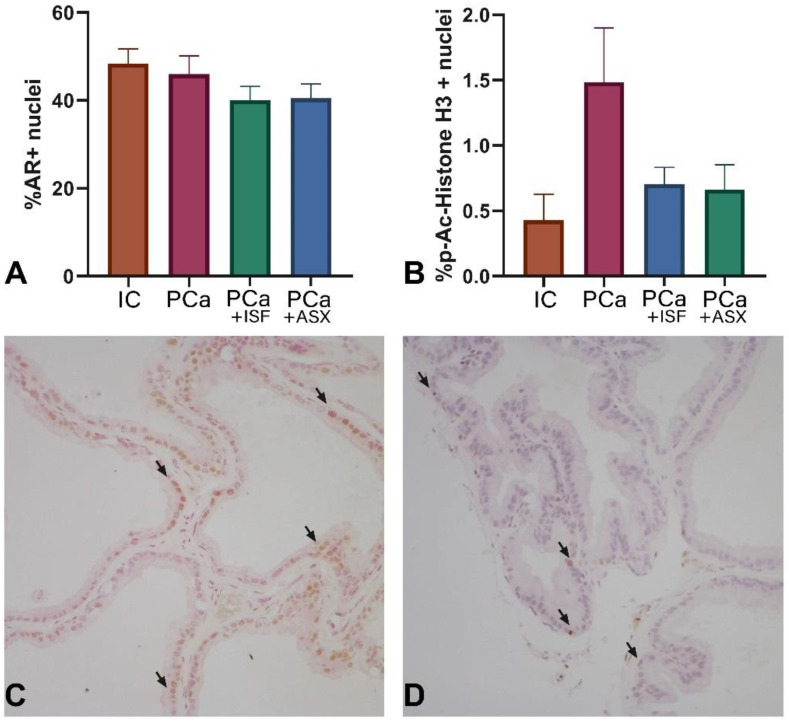
Immunohistochemical staining assessment of prostate ventral lobe for androgen receptor (AR) and p-Ac-Histone H3. IC—intact control. PCa—prostate cancer. ISF—extract of *Pueraria lobata* roots rich in isoflavones. ASX—*Phaffia rhodozyma* extract rich in astaxanthin. (**A**)—percent of AR-positive nuclei cells; (**B**)—percent of p-Ac-Histone H3-positive nuclei cells; (**C**,**D**)—representative microphotographs of AR and p-Ac-Histone H3 immunohistochemical staining in PCa group, correspondingly. Arrows indicate positive nuclei.

**Table 1 plants-12-00564-t001:** Body weight and organ/body weight ratio of rats.

Group	Mean Animal Weight (g)		Relative Organ Weight, mg/100 g			
	At the Start of PCa Induction	At the End of Experiment	Liver	Kidneys	Heart	Spleen
IC	328.9 ± 5.2	537.9 ± 15.2	3207 ± 47	533 ± 12	281 ± 8	218 ± 7
PCa	321.9 ± 5.3	401.4 ± 9.2 ^a^	3603 ± 49 ^a^	830 ± 13 ^a^	344 ± 9 ^a^	302 ± 16 ^a^
PCa + ISF	320.2 ± 5.1	394.9 ± 8.7 ^a^	3622 ± 50 ^a^	810 ± 17 ^a^	336 ± 9 ^a^	326 ± 17 ^a^
PCa + ASX	318.6 ± 4.1	402.3 ± 9.2 ^a^	3652 ± 96 ^a^	832 ± 20 ^a^	343 ± 9 ^a^	288 ± 14 ^a^

IC—intact control. PCa—prostate cancer. ISF—extract of *Pueraria lobata* roots rich in isoflavones. ASX—*Phaffia rhodozyma* extract rich in astaxanthin. ^a^—*q* < 0.05, compared to IC group.

**Table 2 plants-12-00564-t002:** Parameters of prostate carcinogenesis in rats treated with ISF and ASX.

Group	Effective Number of Animals	Number of Animals with Cancer (%)			
		Whole Prostate	Dorsolateral Lobe	Ventral Lobes	Anterior Lobes
IC	31	0 (0%)	0 (0%)	0 (0%)	0 (0%)
PCa	30	11 (37%) ^a^	9 (30%) ^a^	5 (17%) ^a^	0 (0%)
PCa + ISF	27	10 (37%) ^a^	9 (33%) ^a^	4 (15%) ^a^	0 (0%)
PCa + ASX	29	12 (41%) ^a^	9 (31%) ^a^	8 (28%) ^a^	0 (0%)

IC—intact control. PCa—prostate cancer. ISF—extract of *Pueraria lobata* roots rich in isoflavones. ASX—*Phaffia rhodozyma* extract rich in astaxanthin. ^a^—*p* < 0.05, compared to IC group, two-sided Fisher's exact test.

**Table 3 plants-12-00564-t003:** Dorsolateral prostate tissue biochemistry in rats with prostate carcinogenesis treated with ISF and ASX.

Group	SOD, Relative Activity	Ca, µmol/g Protein	Mg, µmol/g Protein	Ca/Mg
IC	100.0 ± 8.1	4.64 ± 0.74	18.4 ± 4.4	0.30 ± 0.07
PCa	71.3 ± 9.1 ^a^	12.92 ± 0.78 ^a^	18.0 ± 2.1	0.76 ± 0.09 ^a^
PCa + ISF	76.3 ± 6.4 ^a^	12.85 ± 0.95 ^a^	16.2 ± 2.8	0.82 ± 0.08 ^a^
PCa + ASX	65.0 ± 6.1 ^a^	8.32 ± 1.62 ^a, b^	17.3 ± 4.7	0.64 ± 0.11 ^a^

IC—intact control. PCa—prostate cancer. ISF—extract of *Pueraria lobata* roots rich in isoflavones. ASX—*Phaffia rhodozyma* extract rich in astaxanthin. SOD—superoxide dismutase. ^a^—*q* < 0.05, compared to IC group; ^b^—*q* < 0.05, compared to PCa group.

**Table 4 plants-12-00564-t004:** Serum biochemistry in rats with prostate carcinogenesis treated with ISF and ASX.

Group	Cholesterol, mmol/L	Calcium, mmol/L	Magnesium, mmol/L	Ca/Mg	Phosphorus, mmol/L	SOD, Relative Activity	MDA, µmol/L^12^ RBC	PSA, ng/mL	Testosterone, ng/mL	DHT, pg/mL
IC	1.07 ± 0.06	2.28 ± 0.04	0.73 ± 0.02	3.14 ± 0.08	1.61 ± 0.04	100.0 ± 1.4	22.0 ± 1.5	1.87±0.15	0.55±0.21	43.6±18.7
PCa	0.93 ± 0.05 ^a^	2.00 ± 0.03 ^a^	0.78 ± 0.01	2.61 ± 0.05 ^a^	1.38 ± 0.05 ^a^	94.6 ± 1.4 ^a^	29.9 ± 2.1	1.65±0.16	11.03±1.08 ^a^	421.5±40.9 ^a^
PCa + ISF	0.76 ±0.04 ^a,b^	2.16 ±0.04 ^a,b^	0.78 ± 0.03	2.84 ±0.08 ^a,b^	1.62 ± 0.06 ^b^	89.0 ±0.9 ^a,b^	38.1 ± 3.6 ^a^	1.28 ±0.11 ^a,b^	16.37±1.05 ^a,b^	547.3±53.9 ^a^
PCa + ASX	0.91 ± 0.03 ^a^	2.11 ±0.06 ^a,b^	0.75 ± 0.03	2.85 ±0.10 ^a,b^	1.49 ± 0.09 ^a^	94.7 ± 2.0 ^a^	31.3 ± 1.5	0.79 ±0.13 ^a,b^	24.96±3.95 ^a,b^	588.6±103.0 ^a,b^

^a^—*q* < 0.05, compared to IC group; ^b^—*q* < 0.05, compared to PCa group; SOD—superoxide dismutase; MDA—malondialdehyde; DHT—5-alpha-dihydrotestosterone.

## Data Availability

Data are contained within the article or Appendix A.

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
