# Peer review of "Effects of Isoflavone-Rich NADES Extract of *Pueraria lobata* Roots and Astaxanthin-Rich *Phaffia rhodozyma* Extract on Prostate Carcinogenesis in Rats"

_plants, 2023, doi:10.3390/plants12030564_

Round 1

Reviewer 1 Report

This paper deals with the evaluation of chemopreventive effect of isoflavones from Pueraria lobata roots extract, rich in isovlavones and Phaffia rhodozyma extract containing astaxanthin in the orthotopic rat model of prostate cancer. In addition, the authors used FTICR-MS technique for annotation of bioactive compounds in the plant extracts.

Moreover, some concerns arise.

The authors claim they used isoflavones and astraxanthin in the PCa model. In fact, P lobata and Ph. rhodozyma extracts, rich in isoflavones and astraxanthin, respectively, were studied.

With this respect, the title is misleading. The plant extracts should be embedded into the title.

The provided scatter plot and histogram were good approach to characterize plant extracts. It’s worth noting that the hyphenated techniques with mass spectrometry result in tentative identification of natural compounds. Accordingly, puerarin, even being in high concentration, was tentatively identified. There no data on the presence of daidzein, genisttein and formononetin in the extract in this study (they have been quantified in the previous investigation, Semenov et al., 2021). In the same manner, the Ph. rhodozyma extract, rich in astraxanthin was used. Consequently, the observed effects could be associated either with additive or synergistic effects of the extracts’ compounds. Thus, the astraxanthin and isoflavons should be omitted as compounds in the animal model and suitable abbreviations of the studied extracts should be embedded.

Why did not the authors attempt to profile the extracts to provide in-depth overview of its metabolome?

The author names of the plants should be introduced at the first mention along with the family name. Keep plant name in  italic through the text.

            It is striking the extremely large number (117) of animals used in the in vivo experiment, divided into only 4 groups of about 30 rats per group. At the same time there is no experimental group treated with a positive control, with a known and established drug in practice. It is not specified how many animals are kept in one cage. For such a long period of time (375 days), it would be good to follow the biochemical parameters in dynamics, or at least at the beginning, before the start of the experiment, in the middle and at the end of the experiment. It would also be useful to follow the effect of at least two doses of the studied extracts to establish a possible dose-response relationship.

Author Response

Answers to Reviewer 1:

Thank you for valuable comments and positive evaluation!

1) The authors claim they used isoflavones and astraxanthin in the PCa model. In fact, P lobata and Ph. rhodozyma extracts, rich in isoflavones and astraxanthin, respectively, were studied. With this respect, the title is misleading. The plant extracts should be embedded into the title.

We have embedded plant extracts in the title of the manuscript:

Lines 1-5 and 24-29 “Isoflavones and astaxanthin in hyperplasia-suppressing doses exert complex effect on rats with induced prostate cancer” changed to “Isoflavone-rich NADES extract of Pueraria lobata roots and astaxanthin-rich Phaffia rhodozyma extract in hyperplas-ia-suppressing doses exert complex effect on rats with induced prostate cancer”

2) The provided scatter plot and histogram were good approach to characterize plant extracts. It’s worth noting that the hyphenated techniques with mass spectrometry result in tentative identification of natural compounds. Accordingly, puerarin, even being in high concentration, was tentatively identified. There no data on the presence of daidzein, genisttein and formononetin in the extract in this study (they have been quantified in the previous investigation, Semenov et al., 2021).

FTICR mass-spectrum of isoflavones was dominated by the signal corresponded to puerarin. Yet, other flavonoids were also found. Daidzien was detected as peak with m/z 253.05067 (error -0.15 ppm), genistein – m/z 269.04556 (error -0.04 ppm) and formononetin – low intensity peak with m/z 267.06592 (error -1.35 ppm). This information was added to the main text:

Lines 83-85: “However, other isoflavones were also detected: daidzien (C15H10O4, m/z 253.05067), genistein (C15H10O5, m/z 269.04556) and formononetin (C16H12O4, low intensity m/z 267.06592).”

In the same manner, the Ph. rhodozyma extract, rich in astraxanthin was used. Consequently, the observed effects could be associated either with additive or synergistic effects of the extracts’ compounds. Thus, the astraxanthin and isoflavons should be omitted as compounds in the animal model and suitable abbreviations of the studied extracts should be embedded.

We have changed designations to extract abbreviations (ISF for NADES extract of Pueraria lobata roots rich in isoflavones and ASX for Phaffia rhodozyma extract rich in astaxanthin) throughout whole manuscript.

3) Why did not the authors attempt to profile the extracts to provide in-depth overview of its metabolome?

We are thankful for this comment. In this research we applied FTICR mass spectrometry. This technique allows to resolve thousands of molecular constituents in a single mixture without hyphenation to HPLC system. Moreover, its analytical capabilities provide opportunity to assign exact elemental composition for the dominance of molecular ions. This is a widely applied approach of untargeted molecular fingerprinting plant metabolites (https://analyticalsciencejournals.onlinelibrary.wiley.com/doi/10.1002/mas.21731).

The disadvantage of this approach is the lack of structural information. Usually for putative metabolite annotation, LC-MS/MS techniques are required, which enable to perform fragmentation of major ions in every chromatographic peak. Yet, this strategy is limited by the current databases, and usually it is possible to annotate dozens of several hundred metabolites in a best-case scenario (https://www.ncbi.nlm.nih.gov/pmc/articles/PMC9366474/). At the same time with FTICR MS we may detect thousands of isobarically-resolved molecular ions. Therefore, for the sake of the current research we believe that fingerprinting is favorable as it provides an overview of samples complexity. We have justified our choice in Discussion section:

Lines 190-195: It is worth noting that the high complexity of the studied samples has been revealed by FTICR-MS applied in an untargeted way, which became an accepted strategy for annotating plant metabolites [13]. Despite the structures of metabolites are missing, the application of FTICR-MS possesses an advantage over conventional LC with tandem mass-spectrometry, which is limited to the current databases, leading to the detection of only dozens of major compounds [14].

4) The author names of the plants should be introduced at the first mention along with the family name. Keep plant name in italic through the text.

Plant names were introduced at the first mention in the manuscript.

5) It is striking the extremely large number (117) of animals used in the in vivo experiment, divided into only 4 groups of about 30 rats per group.

Such numbers per group are needed due to cancer incidence being main assessment index of chemopreventive effect and using lower amount of animals will render experiment to be more susceptible to stochastic variations and able to detect extremely pronounced effects. Compare with Bosland, Maarten C et al. “The MNU Plus Testosterone Rat Model of Prostate Carcinogenesis.” Toxicologic pathology vol. 50,4 (2022): 478-496. (https://www.ncbi.nlm.nih.gov/pmc/articles/PMC9347216/)

At the same time there is no experimental group treated with a positive control, with a known and established drug in practice.

No positive control group was used as there is no currently established in practice drug for prostate cancer prevention. For example, 5α-reductase inhibitors, commonly used for benign hyperplasia treatment and as positive controls for benign hyperplasia models, were not approved by FDA for prostate cancer prevention due to somewhat increased high grade cancer risk (Bosland, Maarten C et al. “A Perspective on Prostate Carcinogenesis and Chemoprevention.” Current pharmacology reports vol. 1,4 (2015): 258-265., https://www.ncbi.nlm.nih.gov/pmc/articles/PMC4591929/).

It is not specified how many animals are kept in one cage.

Animals were kept four per cage, this information was added to Materials and Methods, Lines 305-206 “The animals were kept in conventional polycarbonate cages 1291H (Tecniplast, Italy) with 4 individuals in each under a 12-hour light/dark cycle”

For such a long period of time (375 days), it would be good to follow the biochemical parameters in dynamics, or at least at the beginning, before the start of the experiment, in the middle and at the end of the experiment.

Unfortunately we have analysed only samples obtained at the end of the study, as the main assessed parameter, tumor incidence, was not affected by treatment and so the possible dynamics of biochemical changes could not be linked with the process of prostate carcinogenesis directly.

It would also be useful to follow the effect of at least two doses of the studied extracts to establish a possible dose-response relationship.

Our aim was to test the doses previously shown effective for benign prostatic hyperplasia. Also, the additional groups would require the usage of even more animals in the experiment.

We have made general improvements to manuscript composition and language (Authors changes highlighted in yellow and MDPI English Editing are marked by Microsoft Word markup system).

Reviewer 2 Report

Dear authors,  Your study "Isoflavones and astaxanthin in hyperplasia-suppressing doses exert complex effect on rats with induced prostate cancer" is interesting. However, your introduction, material and methods, results, and discussion sections appear to be similar to other studies as I read your work. MS is poorly written, with numerous grammatical and formatting errors throughout, preventing a thorough understanding of the logic described. Include additional information about the manuscript's uniqueness. I recommend that you correct any grammar and formatting errors and reduce the repetition rate before resubmitting your manuscript for review in this journal.

Author Response

Answers to Reviewer 2:

Dear authors,  Your study "Isoflavones and astaxanthin in hyperplasia-suppressing doses exert complex effect on rats with induced prostate cancer" is interesting. However, your introduction, material and methods, results, and discussion sections appear to be similar to other studies as I read your work. MS is poorly written, with numerous grammatical and formatting errors throughout, preventing a thorough understanding of the logic described. Include additional information about the manuscript's uniqueness. I recommend that you correct any grammar and formatting errors and reduce the repetition rate before resubmitting your manuscript for review in this journal.

Thank you for interest in our work.  We have made general improvements to manuscript composition and language and have submitted our manuscript for the English pre-editing provided by Editorial office (Authors changes highlighted in yellow and MDPI English Editing are marked by Microsoft Word markup system).

We also have added information about the manuscript's uniqueness (Lines 69-71 “To our knowledge, this is the first time isoflavone-rich extract from Pueraria lobata roots and astaxanthin in any form were used in an induced prostate cancer in vivo model.”), expanded details of FTICR MS analysis and improved designation of studied preparations according to comments of Reviewer 1.

Round 2

Reviewer 1 Report

The Authors have addressed my comments from the first round. The manuscript has been improved according to almost all my suggestions. Only some minor points:

l. 84-86 m/z in italic

l.198-201 this statement does not support by the reference 14

Author Response

Thank you again for your valuable comments.

We have put m/z in italics throughout the manuscript.
The reference 14 is corrected and coresponding part of the manuscript improved (lines 191-195).